# Accuracy of Ground Reaction Force and Muscle Activation Prediction in a Child-Adapted Musculoskeletal Model

**DOI:** 10.3390/s22207825

**Published:** 2022-10-14

**Authors:** Kristina Daunoraviciene, Jurgita Ziziene

**Affiliations:** Department of Biomechanical Engineering, Vilnius Gediminas Technical University, LT-10223 Vilnius, Lithuania

**Keywords:** muscular activation, ground reaction force, musculoskeletal model, children gait

## Abstract

(1) Background: Significant advances in digital modelling worldwide have been attributed to the practical application of digital musculoskeletal (MS) models in clinical practice. However, the vast majority of MS models are designed to assess adults’ mobility, and the range suitable for children is very limited. (2) Methods: Seventeen healthy and 4 cerebral palsy (CP) children were recruited for the gait measurements. Surface electromyography (EMG) and ground reaction forces (GRFs) were acquired simultaneously. The MS model of the adult was adapted to the child and simulated in AnyBody. The differences between measured and MS model-estimated GRFs and muscle activations were evaluated using the following methods: the root-mean-square error (RMSE); the Pearson coefficient *r*; statistical parametric mapping (SPM) analysis; the coincidence of muscle activity. (3) Results: For muscle activity, the RMSE ranged from 10.4% to 35.3%, the mismatch varied between 16.4% and 30.5%, and the coincidence ranged between 50.7% and 68.4%; the obtained strong or very strong correlations between the measured and model-calculated GRFs, with RMSE values in the y and z axes ranged from 7.1% to 17.5%. (4) Conclusions: Child-adapted MS model calculated muscle activations and GRFs with sufficient accuracy, so it is suitable for practical use in both healthy children and children with limited mobility.

## 1. Introduction

Movement is a fundamental part of life. It is well-known that human movement offers many advantages, both physically and mentally. It gives healthy joints, strong bones, physical strength, good circulation, good coordination, and reflex reactivity, as well as improved learning skills and concentration. Human movement means freedom to expand oneself through body expression, as well as safety and survival value. Our recent study focuses on the most sensitive group—children. For children, movement is especially important, because it helps them grow. It is a method to establish contact and communication and also can help increase memory, perception, language, attention, emotion, and even decision making [1].

However, only a quality movement allows us to fully enjoy all its benefits. Restriction of movement has an immediate negative effect, so assessing and ensuring the quality of human movement is one of the priorities in science of health and development fields. Studies of gait dynamics are widely conducted by examining the quality of movement of healthy and people with reduced mobility. For both healthy children and those with reduced mobility, the importance of quality of movement is undeniable and influences their future.

The most often examined movement in science is gait. Gait quality is characterized by typical spatiotemporal parameters, kinematic, and kinetic markers. Various motion-recording systems available on the market are not limited to optical cameras, but also to inertial sensors and markerless systems. The following tools are also used in the motion analysis: force plates (Kistler, AMTI, BTS) for GRF measurement, pressure sensor plates or tracks (Zebris, Emed, F-Scan, Pedar, GAITRite, and Tekscan) for evaluating the pressure distribution in the foot, and EMG recording devices (Delsys, Shimmer Research, BTS Free EMG) for quantifying muscle activity. Information from these metrics provides additional evidence needed to process the data and improve the accuracy of musculoskeletal (MS) models. However, the systems also have limitations; for example, measurement of GRF by force plates is one of the most accurate methods, but the following conditions are necessary: a sufficient number of plates to measure as many stance phases as possible during the track; during measurement, the subject has to step only one foot on the plate [2,3]. This condition is more difficult to meet when examining children because of their shorter stride, as children usually step on the plate with both feet. To solve these problems, MS models are capable to calculate GRFs during gait cycles [4,5,6,7].

Significant advances in numerical modelling allow the use of numerical MS models in clinical practice. Most of these models are designed to study adults, very specific neurological conditions, injury risk, or just a few isolated cases [8,9]. MS models based on individual geometry can more accurately estimate individual parameters for each subject [10,11,12,13]. However, the use of individual geometry MS models in clinical practice presents problems of obtaining radiological images. Accordingly, generic geometry MS models are much more convenient, especially for children. It has been observed that a proper disaggregation of such models, taking into account gender and the similarity of anthropometric dimensions, can increase the reliability of the calculation of joint reaction forces [14]. In addition, a multibody model has already been presented in the framework of biotribology of artificial joints of the lower limbs, which confirms the reliability of the joint forces obtained by the models [15]. Recently, scientists are developing a model of generic geometry for children, which would increase the accuracy of the calculated parameters [14]. However, adult models can be applied to children, because no significant differences were observed in children’s models of individual geometry when comparing the generic geometry model of adults [16]. Given that children have different gait parameters than adults, it remains unclear how sensitive generic geometry models are to such changes. Therefore, the application of generic geometry models for the assessment of children’s gait and the correct interpretation of the assessment results of motor functions become particularly relevant. Although highly accurate motion-recording systems and MS models are currently available, there is no modern child gait assessment system that combines these measures in practice.

The lack of an adapted musculoskeletal model for children and the problems related to the difficulties in recording the parameters of children’s movement motivated us to experiment and test the available measures for the analysis of children’s gait. Therefore, the main goal of this work was to use the generic geometry MS model of adults to assess the child’s gait and evaluate its accuracy. The following tasks were involved in achieving the goal: (1) measurement of children’s gait, surface EMGs, and GRFs; (2) adaptation of the adult MS model to a child; (3) MS model accuracy evaluation; (4) testing of the MS model in the case of children with reduced mobility.

## 2. Materials and Methods

### 2.1. Subjects

The accuracy of the model was evaluated with a group of healthy children and simultaneously tested with CP cases. The study involved 17 healthy kids aged 4–11 years (Table 1) and 4 cerebral palsy (CP) children aged 6–8 years (Table 2). Parental consent and child assent were obtained prior to participation in the study.

The criteria for inclusion of typical children in the study were shown as follows: (1) evaluation of the muscle strength of the lower extremities according to the Lovett scale not less than 5 points [17]; (2) ability to understand and follow instructions; (3) absence of motor disorders affecting gait parameters; (4) BMI ≤ 22.9 kg/m^2^ [18,19].

The criteria for inclusion in the CP group were shown as follows: (1) diagnosis of CP; (2) Gross Motor Function Classification System (GMFCS) rating with at least level 3 [20]; (3) ability to walk without a walking aid; (4) ability to understand and follow instructions; (5) ability to walk at least 7 m without stopping.

The criteria for exclusion from both typical and CP children were shown as follows: (1) severe visual impairment; (2) concentration and other significant behavioral disorders.

The experimental protocol was approved by the regional ethical review board (No. 2020/9-1256-738).

For static calibration of the Plug-in-Gait model (described below) and scaling of the MS model, subjects’ anthropometric data were collected: lengths of the thighs, lower legs, feet, and entire legs (from the anterior superior iliac spine to the medial malleolus) and the pelvis, knee, and ankle joints widths. Each subject was given an ID number and filled out a demographic and anthropometric data questionnaire.

### 2.2. Measurement Procedure and Equipment

Motion-recording systems used in the experimental studies: 8-camera (Vero v2.2) optical motion recording system Vicon (Oxford Metrics Group, Oxford, UK, 100 Hz), 10 wireless sensors EMG Delsys Trigno (Delsys, MA, USA, 2000 Hz), and a Bertec 4060-07 (Bertec Corporation, Columbus, OH, USA, 1000 Hz) force plate. Data obtained from different systems with different sampling rates were subjected to pre-processing and normalization.

The study was performed in several stages (Figure 1). Thirty-nine Vicon reflective markers were fixed on the subject’s body comprising the full-body Plug-in-Gait model’s marker set [21]. The detection of gait-events was obtained by a Bertec force plate or toe and heel markers trajectories. The full-body Plug-in-Gait model was scaled according to the individual anthropometric dimensions and later used to pre-process markers trajectories.

A total of 10 EMG sensors were attached to the skin with special disposable double-sided adhesive stickers on five muscles of both legs: biceps femoris (BF), rectus femoris (RF), semitendinosus (SE), lateral gastrocnemius (LG), and medial gastrocnemius (MG).

The start and end of the 7 m distance were clearly marked with sticky tapes on the floor. The child was asked to walk barefoot, at a comfortable speed, from one line to another. Covering a distance of 7 m in one direction was considered as one measurement.

### 2.3. Data Processing

A total of 143 trials of typical children and 32 trials of CP were further processed, equating to an average of 8.9 ± 3.5 trials per healthy child and an average of 8.0 ± 4.7 trials per CP child.

EMG data were pre-processed in the following steps: (1) a best-fit line of the data setup (in the least-squares sense); (2) power spectrum analysis to select the most suitable band pass filter; (3) filtering by a 2nd-order Butterworth band pass filter with cut-off frequencies from 30 Hz to 500 Hz; (4) full-wave rectification; (5) filtering by a low-pass 5th order zero-lag Butterworth filter with a cut-off frequency of 10 Hz; (6) cutting in gait cycles from heel strike to heel strike based on detected gait events. Subsequently, data were normalized in the 0–100% gait cycle. Muscular activations from EMG were obtained by performing the following activities (Figure 2): (i) data transformation using a nonlinear Taeger−Kaiser energy operator (TKEO) [22]; (ii) smoothing; (iii) threshold removing (optimal threshold set up to 35% RMS of data [23]); (iv) normalization of muscle activation to MVC (the MVC value was determined for each subject individually, i.e., the highest value achieved by the subject during all gait cycles); (v) muscle activation onset and offset detection.

The GRF was recorded on three axes (mediolateral *F_x_*, anteroposterior *F_y_*, vertical *F_z_*) in the gait stance phase for both right and left legs. A total of 56 GRF curves from typical children and 10 GRF curves from the CP group were analyzed. GRF data were preprocessed in the following sequence: (1) filtering with a zero-lag 4th-order low-pass Butterworth filter with a 15 Hz cut-off frequency [24,25,26]; (2) cutting from heel strike to toe-off resistance or from the beginning to the end of the stance phase; (3) normalization from 0 to 100% in the stance phase; (4) GRFs normalized to subjects weights to eliminate the effect of different subjects masses.

### 2.4. Model Description and Output

The full-body MS model was first scaled and then used to calculate inverse motion kinematics and dynamics. The simple Plug-in-Gait generic geometry model for adults [22,23] is freely available in AnyBody software (v.7.3, AnyBody Technology A/S, Aalborg, Denmark). It consists of 39 degrees of freedom (DOFs): two spherical hip joints (with 3 DOFs), two inverted knee joints (1 DOF), and two universal ankle joints (2 DOFs) [24]. The AnyBody model was individually adapted for each subject using an anthropometric outline. The same set of whole-body Plug-in-Gait model markers as for the experimental measurements was further used in the model simulations.

The model performs a linear scaling of the segment masses and additionally introduces a specific scaling of the length and mass based on the percentage of body fat tissue [27]. Since the geometry of the model greatly affects the accuracy of the results, the adaptation of the model to children was carried out by the following steps: (1) scaling the model according to the anthropometric values of the subjects; and (2) changing the percentage of the fat layer in adults [24] for children [25].

In the MS model, the muscles are controlled with inverse dynamics by solving the optimization problem through the objective function, and thus, the activations of the selected muscles (BF, RF, SE, LG, and MG) were calculated. The calculation of the GRF is included as part of an algorithm describing muscle function, with 12 points placed on each foot describing muscle-like gears [5]. Five such actuators are added at each contact point, one of which acts as a normal force in the vertical direction of the force plate (z) and the other two pairs of which act in the mediolateral (x) and anteroposterior (y) directions of the force plate and create positive or negative static friction forces. GRFs are calculated for all gait cycles of both legs.

### 2.5. Model Accuracy Evaluation

Model simulation results (i.e., activations from 5 muscles per leg and GRFs from both legs) were further compared with experimental data and tested with CP cases. First of all, the accuracy of the model was evaluated for healthy children to make sure that it is suitable for children. Later, the accuracy of the model was assessed for the pathological gait of children, to check whether the model may function unstable due to the specifics of the gait.

The differences between the GRFs and muscle activation curves from the measurement and the calcuation with the MS model were evaluated using the following methods:(1)the root-mean-square error (RMSE);(2)the Pearson correlation coefficient *r*;(3)statistical parametric mapping (SPM) analysis [28,29];(4)the coincidence of muscle activity (only for a comparison of muscle activations).

In order to evaluate the accuracy of the MS model, the RMSE was calculated as a percentage of the amplitude of the evaluated data. Both RMSEs and mismatch of muscle activations from SPM analysis of less than 20% were evaluated as accurate prediction; accordingly, values ranging from 20% to 50% were considered sufficient accurate; and values of 50% and more were considered inaccurate prediction [30]. Muscular activity coincidence is defined as follows: less than 50% considered as inaccurate prediction; 50–80 percent considered as sufficient accurate prediction; 80% and more considered as accurate prediction.

The absolute values of the Pearson correlation coefficient were interpreted as follows: r ≤ 0.35 represented weak correlation; 0.36 ≤ r ≤ 0.67 represented moderate correlation; 0.68 ≤ r < 0.9 represented strong correlation; r ≥ 0.9 represented very strong correlation [31].

SPM analysis was performed using a paired *t*-test of two independent samples and using the spm1d tool (version M.0.4.7, www.spm1d.org, accessed on 11 September 2022). This analysis made it possible to identify differences in the nature of the curves, i.e., areas where significant differences were found (*p* < 0.01) were indicated. The discrepancy of the curves above the upper threshold indicated that the values of the measured curves were higher than the calculated ones, and the discrepancy below the lower threshold was the opposite—the values of the measured curves were lower than the calculated ones. The results of the muscle activations SPM analysis were presented as the total percentage mismatch of the curves in the gait cycle, i.e., areas, where a statistically significant difference was found and distinguished.

In order to determine the overlap of muscle activity, activations occurring simultaneously in both EMG-measured and MS-calculated curves were evaluated. For this analysis, the curves of the same gait cycles obtained from EMG measurements and MS model calculations were evaluated. The agreement of the muscle activity between the measured and the activation curves calculated by the MS model was evaluated as a percentage (Figure 3). The absence of muscle activity was not assessed in this case.

The normality of distribution of experimental and MS model data was tested by the Lilliefors test for normal distribution [32] at a statistical significance level of *p* < 0.05. Normally distributed data are presented as mean ± standard deviation (SD).

## 3. Results

### 3.1. Accuracy Assessment of Muscular Activation

EMGs and GRFs were measured in all children at 4 to 11 years of age. However, the MS model did not properly simulate 4-year-old participants’ inputs. Therefore, the computational results of this one-child model were not included in the comparison.

First, the muscle activity RMSE values between the experimental and MS model results were evaluated. The RMSE values between healthy and CP subjects mean that the model worked equally reliably with data from both healthy and CP children. RMSE values in the examined muscles ranged, on average, from a minimum of 10.4% to a maximum of 35.3%, and it is not possible to single out the prevailing tendencies of a particular muscle or the subject (the detailed results for each muscle are given in Table 3).

The Pearson coefficients between the experimental EMG and MS model results are given in Figure 4. The values did not visibly differ between healthy and CP subjects, except for the CPd2 case. Again, this means that the model worked equally reliably with the data of both typical and CP children.

The values of the Pearson coefficient ranged from the minimum value of 0.06 to the maximum value of 0.45 on average, i.e., from no to moderate correlations. Higher values outside the interquartile range were observed in healthy children. These outliers arose from a large dataset of different muscle activations in healthy children.

The results of the SPM analysis of healthy children and individual CP cases presented in Figure 5 and Table A1 in Appendix A showed a percentage discrepancy (where *p* is more than the significant level) in muscle activation during the gait cycle.

In general, the mismatch of muscle activations obtained by the MS model ranged from 16.4% to 30.5%. The results were explained in more detail: the mismatches of muscle activations ranged from 17.0% to 31.9% for typical children, from 12.0% to 50.0% for CPh1, from 14.0% to 56.0% for CPd1, and from 0 to 37.0% for CPd3. All activations for CPd2 coincided. The mean mismatch of activations varied between different muscles, but the largest mismatch was observed in the muscle activations of BF, SE, and MG for the CPh1 case and in BF and SE muscles for the CPd1 case. It should be noted that there was no discrepancy in muscle activation of the CPd2 subject, and the mean mismatch of the CPd3 subject did not exceed the 26% limit.

Figure 6 and Table A2 in Appendix A show the results of the activity overlap between the curves. The average coincidences of all muscle activity obtained in the MS model ranged from 50.7% to 68.4%, from 58.1% to 75.6% for typical children, from 47.3% to 75.8% for the CPh1 case, from 48.4% to 72.7% for the CPd1 case, from 38.1% to 92.9% for the CPd2 case, and from 32.6% to 75.5% for the CPd3 case. No difference was observed between healthy and CP subjects.

### 3.2. Accuracy Assessment of GRFs

The normalized GRF curves (*F_nx_*, *F_ny_*, and *F_nz_*) of the measurements and the MS model predictions were compared (Figure 7 and Figure 8). In this assessment, CP subjects were grouped together due to the small sample of the measured GRF curves. Analyzing the correlation of vertical (*F_nz_*) and anteroposterior (*F_ny_*) forces (measured vs. model-predicted curves) in healthy children, a very strong correlation was observed on both sides of the body (Figure 7a and Figure 8a), i.e., values from 0.92 to 0.96. However, an average correlation was found between the mediolateral (*F_nx_*) forces on both sides of the body, i.e., from 0.54 to 0.67. Analyzing the correlation of the vertical (*F_nz_*) and anteroposterior (*F_ny_*) forces in the CP group, a very strong (0.92) correlation was observed on the right side of the body (Figure 7c), while on the left (Figure 8c) a strong (0.76–0.88) correlation was found. However, a strong correlation (0.76) was found between the mediolateral (*F_nx_*) forces on the right (Figure 7c), and the mean correlation (0.56) was identified on the left (Figure 8c).

RMSE values in healthy and CP children were similar. The RMSEs for all children were between 37.0% and 97.8% for the mediolateral (*F_nx_*) forces, between 7.9% and 17.5% for the anteroposterior (*F_ny_*) forces, and between 7.1% and 10.9% for the vertical (*F_nz_*) forces. The results showed that the anteroposterior (*F_ny_*) and vertical (*F_nz_*) forces determined by the MS model were accurate and the mediolateral (*F_nx_*) forces were sufficiently accurate (except for the right side of HCH). The results of the SPM analysis in healthy children showed discrepancies in anteroposterior (*F_ny_*) and vertical (*F_nz_*) forces on both sides of the body (Figure 7b and Figure 8b). The measured values of the deceleration peaks of the right anteroposterior (*F_ny_*) force were higher calculated by the MS model (Figure 7b), and the measured values of the left side (higher than the calculated ones) did not affect the gait evaluation because the mismatch did not cover the deceleration peak area (Figure 8b). The mismatch of the values of the anteroposterior (*F_ny_*) force curves on both sides of the body in the middle of the gait cycle also did not affect the gait assessment, as there were no commonly assessed force peaks in this zone. Lower values of the measured vertical force (*F_nz_*) were observed in the middle of the gait cycle. When evaluating GRF parameters during gait, these discrepancies may affect the interpretation of the mid-force of the support phase.

In the CP cases, only right anteroposterior (*F_ny_*) and vertical (*F_nz_*) force mismatches were observed (Figure 7d). These disparities did not affect the evaluation, because the misalignment of the anteroposterior (*F_ny_*) force did not fall within the area of the peaks to be evaluated and the area of misalignment of the vertical (*F_nz_*) force was very small. No GRFs’ mismatches were observed on the left side of the CP group (Figure 8d).

## 4. Discussion

Musculoskeletal models are increasingly being used to analyze and evaluate movement quality, and their advantages and benefits are becoming increasingly apparent.

Unfortunately, there are many challenges to using these models outside of their original purpose. Our work aimed to verify whether the general-geometry MS model of an adult can obtain accurate results when it is adapted to a child and be applied in practice.

The obtained results of the accuracy and stability of muscle activations in the MS model showed what activation overlap can be expected. The differences between the measured and calculated muscle activations resulted from the following reasons:MS model activation start delay [33];The time delay introduced by a low-pass Butterworth digital filter selected for EMG data [34];Absence of a paediatric pattern of muscle activation in the gait cycle, i.e., muscle activation in each child should be assessed individually [35,36];A large number of the MS model outputs, because a particular muscle is divided into several fibres, and during the measurements the EMG signal is more related to the activity in the largest parts of the muscle closest to the electrodes [37].

In addition, even other researchers who conducted studies evaluating the accuracy of this MS model of adults [37] and other MS models [14,38,39,40] have observed similar trends: the MS model reflects the most important features of experimental EMG data, although the activation times and amplitudes of some muscles differ when compared with the EMG data. Another factor in the discrepancy is the fact that it is impossible to accurately measure muscle force and activation, and some muscles cannot be studied experimentally at all. This makes MS models particularly important, because in many cases they are the only way to estimate certain valuable information, such as internal body forces.

Based on the estimation of RMSE accuracy provided by other scientists [30] and our insights, we defined the accuracy of our obtained muscle activation results as accurate and sufficiently accurate: RMSE from 10.4% to 35.3%; mismatch of muscle activations from 16.4% to 30.5%; and muscular activity coincidence from 50.7% to 68.4%. The Pearson coefficients between experimental EMG and MS model results revealed from no to moderate relationships due to the presence of the activation delay of the MS model.

In addition, in our study, the MS model accurately calculated the forces *F_y_* and *F_z_*, and less reliable results were obtained by calculating the forces *F_x_*. Therefore, the calculation of GRF force in all gait cycles solved the problems related to GRF measurements in children, when it was difficult for children to step correctly on the force plates, which increased the number of repetitions of measurements, changed the pattern of natural gait, and so on.

Notably, researchers studying the accuracy of GRFs determination in adult’s MS model have obtained very similar results [5]: a 0.96 correlation for the vertical and anteroposterior GRFs compared to our correlation results of 0.92–0.96 for HCH and 0.76–0.96 for CP); a 0.81 correlation for the mediolateral GRF compared to our correlation results of 0.54–0.76 for HCH and 0.56–0.67 for CP.

The mismatches of muscular activations from 16.4% to 30.5%, the coincidence of muscle activities from 50.7% to 68.4%, and the strong or very strong correlations of the calculated GRFs and the RMSE values between the GRF values from 7.1% to 50% allowed us to state that the child-adapted MS model was accurate and stable and the following advantages should be highlighted: (1) it allows reducing the number of expensive measurement techniques for measuring and analysing movement; (2) it saves time for staff to prepare test and measurement techniques and subsequently process and analyze information; (3) children in the study do not need to put on additional sensors and be instructed on how to walk to step the force plates; (4) it provides more information (more muscle can be analyzed) than what can be measured in all gait cycles; and (5) the quality of motion analysis significantly improves what leads to a more accurate selection of the most optimal innervations or compensatory mechanisms for people with movement and workouts or exercises for healthy children.

We believe that research in this direction should be continued, but the most important limitations of this work should be taken into account. A wide-age group of children was sampled and evaluated. However, we believe that differences in the model inputs between children of different ages may vary and influence the output results of the MS model, as do differences in BMI. This may be due to larger scaling errors in the overall adult MS model because of large differences in anthropometry between children and adults [13,38]. The case of each child, both healthy and disabled, must be assessed individually and even linked to the results of other movement assessments if necessary.

Our further work is aimed at potentially larger samples of children with limited mobility (as CP). Their experimental gait studies and the application of the model in clinical practice would be aimed at developing a tool that would help to accurately identify pathologies and limitations, select the most optimal interventions and develop rehabilitation protocols. Another relevant direction of the science is the modification and improvement of the model, taking into account the limitations we have already known.

## 5. Conclusions

The results of the model accuracy assessment showed that the child-adapted MS model calculated muscle activations and GRFs with sufficient accuracy, making it suitable for use in practice in both healthy children and those with reduced mobility for movement analysis and evaluation. In addition, the use of the MS model solves many problems associated with experimental research in children, making it very convenient and time-saving.

## Figures and Tables

**Figure 1 sensors-22-07825-f001:**
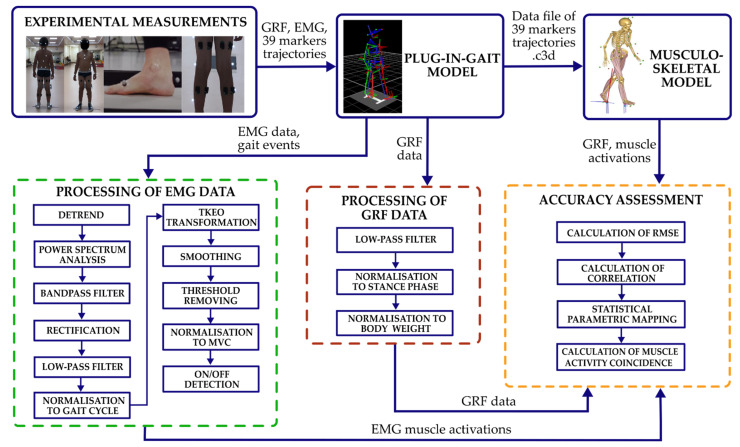
Procedures to conduct the study.

**Figure 2 sensors-22-07825-f002:**
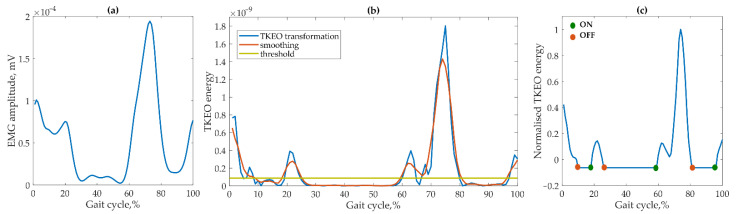
Stages of EMG data processing: (**a**) pre-processed EMG; (**b**) TKEO transformation, smoothing, and threshold determination; (**c**) normalization to MVC and onset/offset detection.

**Figure 3 sensors-22-07825-f003:**
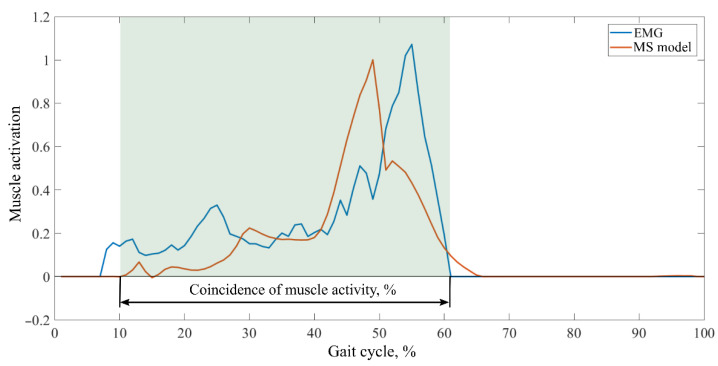
Assessment of the coincidence of the muscle activity duration.

**Figure 4 sensors-22-07825-f004:**
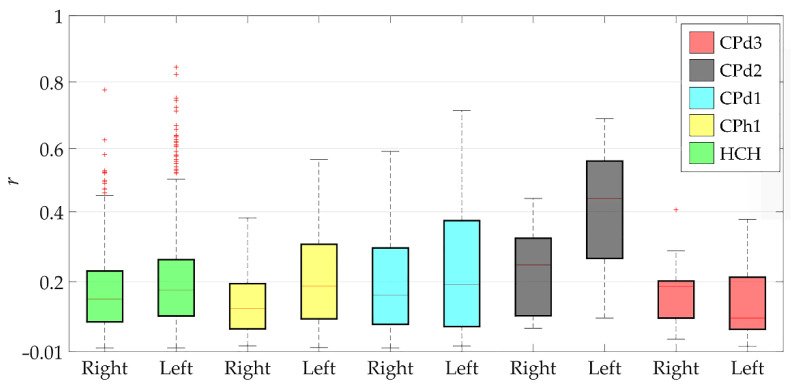
Mean values of the correlation coefficients. +, outliers of boxplots; HCH, healthy children.

**Figure 5 sensors-22-07825-f005:**
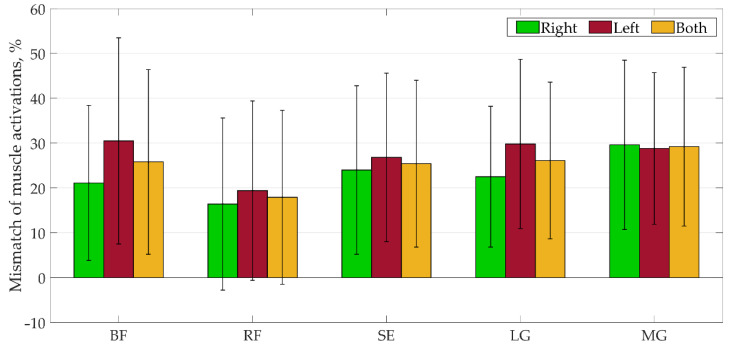
Results of the SPM analysis for all subjects. Data are presented as mean ± SD.

**Figure 6 sensors-22-07825-f006:**
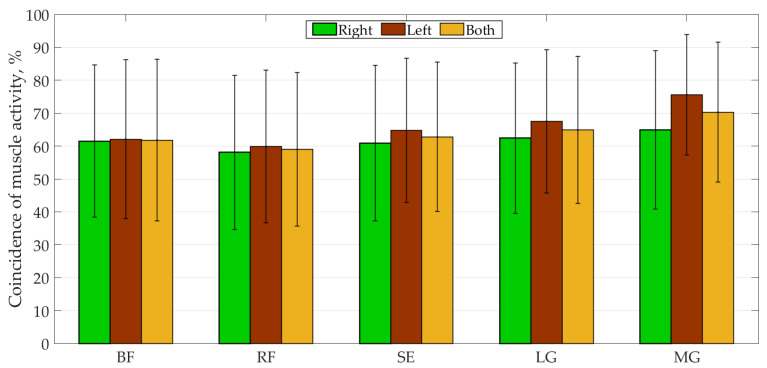
Coincidence of all muscle activities for all subjects. Data are presented as mean ± SD.

**Figure 7 sensors-22-07825-f007:**
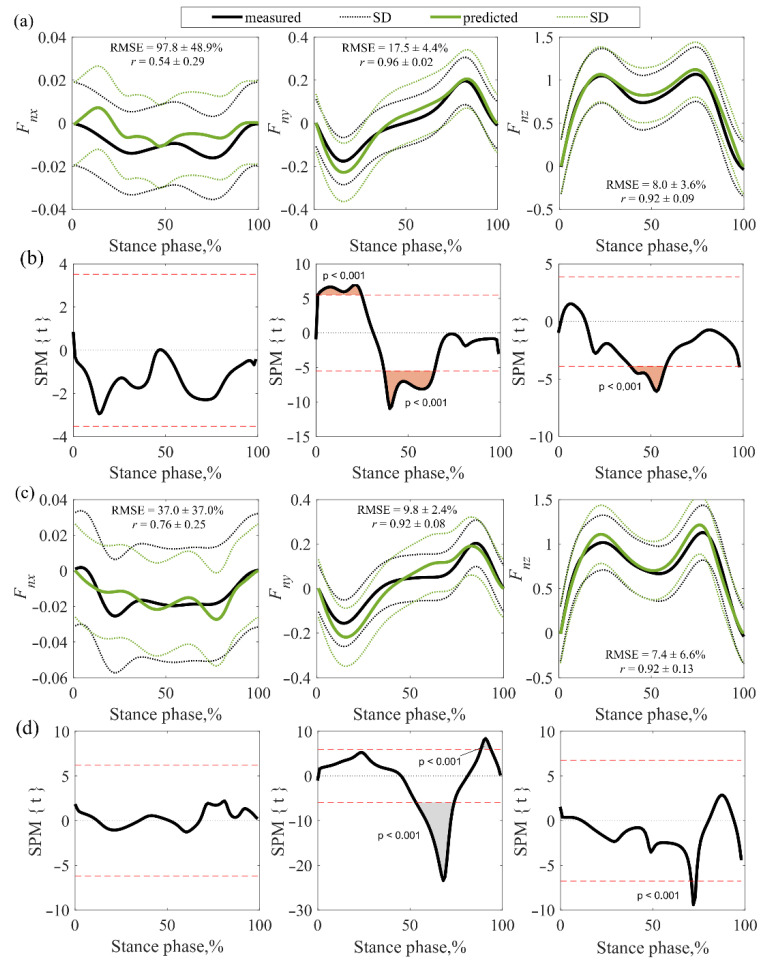
Results of the comparison of the measured and model predicted GRFs for the right leg: (**a**) averaged force curves for the HCH; (**b**) mismatch of HCH curves; (**c**) averaged force curves for the CP subjects; (**d**) mismatch of CP curves.

**Figure 8 sensors-22-07825-f008:**
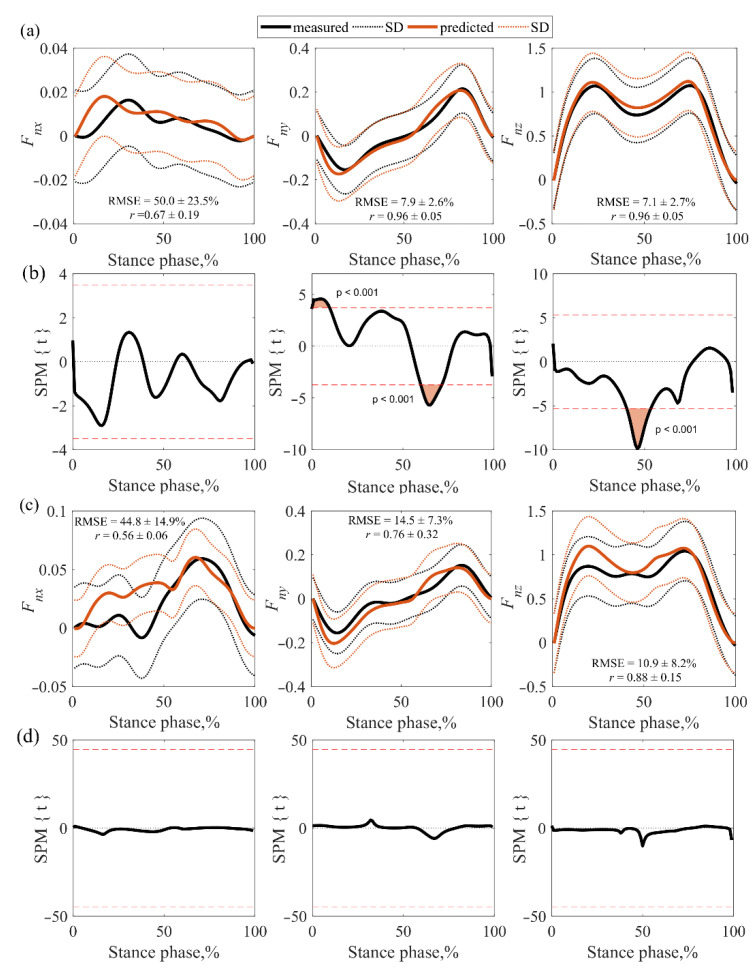
Results of the comparison of the measured and model predicted GRFs for the left leg: (**a**) averaged force curves for the HCH; (**b**) mismatch of HCH curves; (**c**) averaged force curves for the CP subjects; (**d**) mismatch of CP curves.

**Table 1 sensors-22-07825-t001:** Demographic and descriptive data of the participants (n = 17).

	Age (year)	Height (m)	BMI (kg/m^2^)
Male (n = 6)	9.0 ± 1.6	1.340 ± 0.109	15.9 ± 1.8
Female (n = 11)	7.6 ± 2.0	1.299 ± 0.108	16.6 ± 2.4
All	7.9 ± 2.0	1.314 ± 0.107	16.4 ± 2.2

Data are presented as mean ± SD.

**Table 2 sensors-22-07825-t002:** Demographic and descriptive data of CP subjects (n = 4).

Marking	CP Form	GMFCS Score	Affected Side	Gender	Age (year)	Height (m)	BMI (kg/m^2^)
CPh1 (n = 1)	H	1	right	F	7.0	1.320	17.9
CPd1 (n = 1)	D	1	both	F	8.0	1.370	16.4
CPd2 (n = 1)	D	2	both	M	7.0	1.270	13.6
CPd3 (n = 1)	D	3	both	M	6.0	1.180	15.8
All	-	1.6 ± 0.9	-	-	7.0 ± 0.82	1.285 ± 0.08	15.4 ± 1.8

Data are presented as mean ± SD. H, hemiplegic; D, diplegic; M, male; F, female.

**Table 3 sensors-22-07825-t003:** RMSEs between muscle activations.

Subjects	Body Side and Stride Number	RMSE (%)				
		BF	RF	SE	LG	MG
HCH	Right (n = 376)	21.4 ± 9.3	22.0 ± 6.4	26.1 ± 9.2	22.4 ± 2.3	25.4 ± 5.4
	Left (n = 373)	26.0 ± 9.4	20.5 ± 7.9	27.1 ± 6.8	21.1 ± 6.2	25.5 ± 7.8
CPh1	Right n = 41)	23.0 ± 7.6	18.4 ± 7.6	22.2 ± 6.8	22.8 ± 6.3	24.7 ± 6.1
	Left (n = 41)	29.1 ± 9.1	18.3 ± 6.4	29.3 ± 6.2	20.2 ± 5.0	22.8 ± 5.7
CPd1	Right (n = 32)	30.7 ± 4.2	20.2 ± 6.5	32.1 ± 7.3	21.8 ± 12.0	14.8 ± 4.5
	Left (n = 32)	30.8 ± 6.4	16.9 ± 13.5	24.9 ± 4.5	14.5 ± 8.9	16.5 ± 9.5
CPd2	Right (n = 2)	21.4 ± 5.5	13.8 ± 7.1	14.5 ± 10.1	25.4 ± 6.7	27.9 ± 8.5
	Left (n = 1)	18.1	10.4	22.7	21.9	28.1
CPd3	Right (n = 13)	18.9 ± 7.3	32.8 ± 7.1	18.4 ± 9.2	34.3 ± 8.4	35.2 ± 9.4
	Left (n = 13)	18.7 ± 5.9	32.6 ± 9.7	14.7 ± 8.9	35.3 ± 8.7	33.5 ± 9.0

Data are presented as mean ± SD. H, hemiplegic; D, diplegic; M, male; F, female.

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
