# Peer review of "Accuracy of Ground Reaction Force and Muscle Activation Prediction in a Child-Adapted Musculoskeletal Model"

_sensors, 2022, doi:10.3390/s22207825_

Round 1

Reviewer 1 Report

This paper was aimed to propose a Musculoskeletal Model derived from an adapted MS model for adults, by evaluating also its accuracy.

The Authors followed they objective by

1)measuring the children's gait;

2) acquiring simultaneously surface electromyography (EMG) and ground reaction forces (GRFs);

3) adapting the adult MS model to a child by scaling the model according to the anthropometric values of the subjects and by changing the percentage of the fat layer in adults for children;

4) evaluating the model accuracy and testing the obtained model for children with reduced mobility (disabled).

The paper is well organized and built through the above four points, treating in rigorous and readable way the topic object of investigation, and achieving novel and interesting results and conclusions.

I have only minor suggestion for the authors:

-to improve the scientific framework for taking not account also recent and relevant paper on the modeling of MS systems;

-to highlight in the conclusions the limitation of the investigations, depicting future developments of the proposed research.

Author Response

First of all, we thank the reviewer for his highly positive evaluation of our scientific publication and for pointing out the missing information in the manuscript.

Point 1: to improve the scientific framework for taking not account also recent and relevant paper on the modeling of MS systems;

Response 1:  If we have understood the comment well, we believe and strive to create new knowledge through our scientific insights and studies, while also relying on the latest achievements in science in this field. In addition, we rely on practical experience in cooperation with child rehabilitation specialists in our country.

In response to the reviewer's comment, we slightly strengthened the introduction part and cited several more scientific works indirectly contributing to our knowledge.

Reference list updated by new sources: [8], [9], [15].

Point 2: to highlight in the conclusions the limitation of the investigations, depicting future developments of the proposed research.

Response 2:  We thank the reviewer for the comment and accept it. We indicated our limitations at the end of the discussion (in the last paragraph), but we strengthened it with future developments and a description of activities.

Reviewer 2 Report

The paper presents an interesting investigation regarding the accuracy of model predicted muscles activation and ground reaction forces in children’ gait analysis.

Several methods are proposed to evaluate the differences between measured and simulated values.

In par 2.1 subject selection criteria are presented. In general they are based on literature or on practical considerations, but the BMI limit is not justified. Was it defined by the authors? Is there any correlation between results and subject’s anthropometric parameters (such a s BMI)?

When presenting results tables give a large set of values rather difficult to be interpreted at a first read. I suggest to use a graphical form, as done for correlation coefficients in figure 4, where box plots gives to the reader immediately a clear picture of the results. Tables give of course further detail and they can be left where they are, or moved in an appendix.

When commenting results, some more explanations could be useful for the reader for example: why the model is considered to perform stably (line 239 and other)? Authors are comparing a mean RMSE on a set of healthy subjects and single RMSE for single specific CP subjects. Beside that the RMS range is rather wide: 10-35%.

At line 256 a correlation from 0.06 to 0.45 is considered to range from weak to strong. How is this justifiable? I would say a from no to a poor correlation. Moreover this is in contrast with strong definition at line 208.

Results outside the interquartile range can not be excluded because of sample numerosity. Please reconsider their exclusion, or motivate it robustly.

Please revise carefully the overall text, for example lines 219 and 228 are the same, in table  4 lines for CDP2 can be removed.

Author Response

Response to Reviewer 2 Comments

We thank the reviewer for his interest in this topic and kind reflection. We made improvements to the manuscript based on the reviewer's comments.

We hope and trust that all reviewers' comments and concerns have been fully and satisfactorily addressed. Detailed responses to comments follow.

Point 1: In par 2.1 subject selection criteria are presented. In general they are based on literature or on practical considerations, but the BMI limit is not justified. Was it defined by the authors? Is there any correlation between results and subject’s anthropometric parameters (such as BMI)?

Response 1:  We thank the reviewer for directing our attention to the missing information in the manuscript regarding BMI limits. BMI cut-offs were determined based on results reported in the literature and practical observations.

Children's movements during gait may be unstable due to insufficiently developed posture and gait control, so they are characterized by quite high variability of gait parameters between different individuals and between individual steps of the same child (Hausdorff et al., 1999). This may be due to the different development of neural control between children, which directly determines also motor control, or individual anthropometric indicators, such as excess weight or height (Dewolf et al., 2020; Dufek et al., 2012; Nantel et al., 2006). As the child grows, these changes gradually decrease,  i.e. approaching the parameters characteristic of adult gait.

References:

Hausdorff, J. M., Zemany, L., Peng, C. K., & Goldberger, A. L. (1999). Maturation of gait dynamics: Stride-to-stride variability and its temporal organization in children. Journal of Applied Physiology, 86(3), 1040–1047.

Dewolf, A. H., Sylos-Labini, F., Cappellini, G., Lacquaniti, F., & Ivanenko, Y. (2020). Emergence of Different Gaits in Infancy: Relationship Between Developing Neural Circuitries and Changing Biomechanics. Frontiers in Bioengineering and Biotechnology, 8, 473.

Dufek, J. S., Currie, R. L., Gouws, P. L., Candela, L., Gutierrez, A. P., Mercer, J. A., & Putney, L. A. G. (2012). Effects of overweight and obesity on walking characteristics in adolescents. Human Movement Science, 31(4), 897–906.

Nantel, J., Brochu, M., & Prince, F. (2006). Locomotor Strategies in Obese and Non-obese Children. Obesity, 14(10), 1789–1794.

We have cited literature sources at the BMI cut-offs indicated. Based on the literature and practical observations of clinicians, we can see that BMI does have an impact on gait parameters and possible model output results. We did not investigate correlations with BMI in this work, but we noted that this would also be useful material for optimizing model errors. At the end of the discussion, we mentioned this aspect to the stated limitations of the study.

Point 2: When presenting results tables give a large set of values rather difficult to be interpreted at a first read. I suggest to use a graphical form, as done for correlation coefficients in figure 4, where box plots gives to the reader immediately a clear picture of the results. Tables give of course further detail and they can be left where they are, or moved in an appendix.

Response 2:  In response to a reviewer's comment, we graphically represented the core of Tables 4 and 5 (resulting in Figures 5 and 6, respectively), and moved the rest of the quantitative information to the appendices (see Appendix Tables 1 and 2). We hope that it will now be much clearer for the reader to follow the results.

Point 3: When commenting results, some more explanations could be useful for the reader for example: why the model is considered to perform stably (line 239 and other)? Authors are comparing a mean RMSE on a set of healthy subjects and single RMSE for single specific CP subjects. Beside that the RMS range is rather wide: 10-35%.

Response 3:  Maybe that sentence with the term "stable" is a bit unclear, so we corrected it (in the Results section marked in red).

 We wanted to say that the model works reliably both with data from healthy children and with CP cases, that is, it does not hang, does not stuck or does not break, and gives results within the same limits.

According to these recommendations accordingly, we defined the threshold to 20 % is the accurate prediction and above 50 % - inaccurate.

We explained and substantiated the accuracy limits of our model in the discussion section, where this literature source was cited (marked in red).

*Reference

Halawi L, Clarke A, George K. Evaluating Predictive Performance. Harnessing Power Anal 2022; 51–59.

Point 4: At line 256 a correlation from 0.06 to 0.45 is considered to range from weak to strong. How is this justifiable? I would say a from no to a poor correlation. Moreover this is in contrast with strong definition at line 208.

Response 4:  We thank the reviewer for directing our attention to the confusing information in the manuscript regarding correlations. We made a mistake in the description of the results. Based on our correlation bounds in the methodological section in subsection 2.5 (*), we corrected it to " The values of the Pearson coefficient range from minimum 0.06 to maximum 0.45 on average, i.e., from no to moderate correlation ".

*The absolute values of the Pearson correlation coefficient are interpreted as follows: r ≤ 0.35 – weak correlation, 0.36 ≤ r ≤ 0.67 – moderate correlation, 0.68 ≤ r < 0.9 – strong correlation, r ≥ 0.9 – very strong correlation [29].

Point 5: Results outside the interquartile range can not be excluded because of sample numerosity. Please reconsider their exclusion, or motivate it robustly.

Response 5: 

While trying to understand the reviewer's comment, we also found our own sentence (“These values are due to the large sample of healthy children and are not considered in the analysis of the mean values for all muscles”) in the results section, which most likely caused confusion. This sentence is incorrect.

We did not exclude outliers in the overall assessment of muscle activity in healthy children. In Figure 4, outliers were not rejected, they were estimated.

We rephrased our statement and made sure that it would not be repeated in the article.

Point 6: Please revise carefully the overall text, for example lines 219 and 228 are the same, in table  4 lines for CDP2 can be removed.

Response 6:  We accept the reviewer's suggestion and revised the overall text carefully. 

* The null values in table 4 for case CP2 are not an error, but show the total percentage mismatch of the curves in the gait cycle, i.e., there are no areas where a statistically significant difference is found.
